# IL-36 Cytokines: Regulators of Inflammatory Responses and Their Emerging Role in Immunology of Reproduction

**DOI:** 10.3390/ijms20071649

**Published:** 2019-04-03

**Authors:** José Martin Murrieta-Coxca, Sandra Rodríguez-Martínez, Mario Eugenio Cancino-Diaz, Udo R. Markert, Rodolfo R. Favaro, Diana M. Morales-Prieto

**Affiliations:** 1Placenta Lab, Department of Obstetrics, Jena University Hospital, 07740 Jena, Germany; josemartin.murrietacoxca@uni-jena.de (J.M.M.-C.); rodolfo.favaro@med.uni-jena.de (R.R.F.); diana.morales@med.uni-jena.de (D.M.M.-P.); 2Departamento de Inmunología, Instituto Politécnico Nacional, Escuela Nacional de Ciencias Biológicas, 11340 Mexico City, Mexico; sandrarodm@yahoo.com.mx (S.R.-M.); mecancinod@gmail.com (M.E.C.-D.)

**Keywords:** IL-1 superfamily, IL-36 cytokines, pregnancy, uterus, inflammation

## Abstract

The IL-36 subfamily of cytokines has been recently described as part of the IL-1 superfamily. It comprises three pro-inflammatory agonists (IL-36α, IL-36β, and IL-36γ), their receptor (IL-36R), and one antagonist (IL-36Ra). Although expressed in a variety of cells, the biological relevance of IL-36 cytokines is most evident in the communication between epithelial cells, dendritic cells, and neutrophils, which constitute the common triad responsible for the initiation, maintenance, and expansion of inflammation. The immunological role of IL-36 cytokines was initially described in studies of psoriasis, but novel evidence demonstrates their involvement in further immune and inflammatory processes in physiological and pathological situations. Preliminary studies have reported a dynamic expression of IL-36 cytokines in the female reproductive tract throughout the menstrual cycle, as well as their association with the production of immune mediators and cellular recruitment in the vaginal microenvironment contributing to host defense. In pregnancy, alteration of the placental IL-36 axis has been reported upon infection and pre-eclampsia suggesting its pivotal role in the regulation of maternal immune responses. In this review, we summarize current knowledge regarding the regulatory mechanisms and biological actions of IL-36 cytokines, their participation in different inflammatory conditions, and the emerging data on their potential role in normal and complicated pregnancies.

## 1. Introduction

Pregnancy requires a unique immune program in which the conceptus must be tolerated and supported by the maternal organism, even though half of its genetic characteristics derive from the father. Professor Peter B. Medawar was the first to notice that, in immunological terms, the fetus has a similar condition as a semi-allogenic graft. This observation was based on the assumption that the placenta expresses paternal antigens and, therefore, under normal immunological conditions, should be rejected [1]. He proposed that there is no response against the fetus because the placenta represents a functional and anatomical barrier that isolates the fetal components from the maternal immune response [2]. To prevent rejection, the immune system enters into a dynamic and cooperative state that modulates the inflammatory responses allowing the correct development of the fundamental events of pregnancy [3,4]. Although numerous immunological processes in pregnancy and the placenta have been studied and described, many immune mechanisms acting during the normal development of pregnancy or the menstrual cycle are still not fully understood. Several molecules such as cytokines, growth factors and hormones actively participate in these processes. Together, they modulate a non-pathological form of inflammation in pivotal stages such as endometrial decidualization, embryo implantation, and induction of childbirth.

### 1.1. The Immune System and Pregnancy

Immune tolerance at the maternofetal interface is an intrinsic event by which the maternal immune system does not initiate an aggressive response against the fetus. Trophoblast cells and the maternal immune system develop and establish a cooperative state, helping each other to attain a successful pregnancy [5]. The immune tolerance of pregnancy comprises a sequence of events and interactions from conception, implantation and placentation until labor and delivery [6]. Currently, several cellular and molecular mechanisms that aid in the modulation of maternal immune responses against paternal antigens have been identified [7,8]. Trophoblast cells establish a cross-talk with T cells through soluble mediators and exosomes [9], leading to the generation and activation of regulatory T cells (Tregs) [10]. Factors carried by extracellular vesicles “educate” resident and newly recruited Treg cells that suppress Th1 cell responses, favoring fetal tolerance [11].

The immune system protects the mother against environmental threats and prevents harm to the fetus. During pregnancy, the maternal immune system presents a strengthened network of recognition, communication, trafficking and repair. Cells and factors of fetal origin such as IL-1, TNF-α, MCP-1, MIP1-β, and IFN-γ modify maternal responses to stimuli [12,13]. These fetal proinflammatory cytokines sensitize the pregnant mother to bacterial products and promote preterm labor [12]. When required, alarm and defensive responses can be mounted in order to maintain the welfare of the mother and fetus. Therefore, the maternal immune system is not suppressed as previously assumed. On the contrary, it is active, functional, and carefully controlled [14]. It has been proposed that maternal immune interactions with fetal cells involve three major stages: a) attraction of trophoblast cells and recruitment of chemokine-secreting immune cells to the implantation site [5,15]; b) education of trophoblast cells to produce regulatory cytokines that modulate the immune cell differentiation process [16], and c) the response of these trophoblast-instructed immune cells, which react to local microenvironmental signals in a unique way [17].

The endometrium of both humans and mice is populated by a rich and dynamic variety of immune cells. Regarding early human pregnancy, around 50–75% of the decidual cells are CD45+ leukocytes [18,19]. Of these leukocytes, 65–70% are uterine natural killer (NK) cells, 10–20% are antigen presenting cells (APCs) such as macrophages, and 2–4% are dendritic cells (DCs) [15,20,21]. About 20% of the lymphocytes in the uterus are represented by T cells, from which 1% is composed by Tregs [22]. In contrast to peripheral blood, where CD4+ T cells form the predominant T cell subset, in decidual tissue at term pregnancy CD8+ T cells are the most abundant T cell subset. The majority of these CD8+ T cells are activated effector memory T cells. Unprimed naive cells are virtually absent [23]. A small proportion of the T cells found in early pregnancy are γδ T cells and constitute approximately 30% of decidual CD3+ T cells [24,25]. Likewise, in contrast to cytotoxic peripheral blood NK cells, uterine NK cells present a unique immunomodulatory phenotype characterized by the secretion of an assorted repertoire of cytokines and angiogenic factors. These cells modulate trophoblast invasion and spiral artery remodeling at the maternofetal interface [26,27]. An imbalance in the number and functionality of uterine NK cells and endothelial progenitors together with changes in pro- and anti-angiogenic factors contributes to preeclampsia onset [28].

DCs and APCs are heterogeneous populations of cells that initiate and coordinate the innate and adaptive immune response. These cells accumulate in the uterus prior to implantation and remain throughout pregnancy [29,30]. APCs appear to play a central role in the configuration of the cytokine profile at the maternofetal interface [29,31]. However, decidual DCs do not contribute to the presentation of embryo-derived antigens in the uterine lymph nodes. Instead, they remain trapped in the pregnant uterus, even when induced to mature by a lipopolysaccharide (LPS) stimulus [32]. This immobilization inhibits a major immune threat to fetal survival, as resident tissue DCs are crucial initiators of the immune response executed by T cells [33,34]. Little is known regarding the role of neutrophils in human reproduction. Recent studies suggest that under the influence of the decidual microenvironment, neutrophils adopt a unique phenotype different from peripheral blood to release pro-angiogenic factors [35].

The juxtaposition of fetal structures, particularly trophoblast cells, and the decidualized endometrium gives rise to the maternofetal interface. At this site, trophoblast cells come into contact with different endometrial leukocytes and stromal cells, modulating their immune responses in a mutual way [36,37]. The dynamic balance between immunological, cellular and molecular elements is essential for the correct development of pregnancy. According to the prevalence of pro-inflammatory or anti-inflammatory factors, distinct immunological stages are distinguished during pregnancy.

### 1.2. Pro-Inflammatory and Anti-Inflammatory Responses during Pregnancy

The presence of immune cells at the maternofetal interface is not associated with a response against the fetus, in fact these cells are attracted to protect and facilitate pregnancy. A challenging question is when and why the protective and supportive maternal immune system becomes a foe in pathological pregnancies.

Pregnancy has three immunological phases characterized by distinct biological processes [38,39]. The initial phase involves embryo implantation and early stages of placentation, encompassing the first and the beginning of the second trimester of pregnancy. This period resembles an ‘open wound’, leading to a strong inflammatory response mainly due to implantation, invasion and the vascularization of trophoblast cells into the maternal endometrium [40]. These events create a ‘battlefield’ between immune, invasive, dead, and repairing cells [15,41]. Therefore, an inflammatory environment is mandatory in order to ensure an adequate reconstruction of the uterine epithelium, elimination of cellular debris, and tissue remodeling. In this context, the first trimester of pregnancy is considered as a pro-inflammatory phase in which Th1-type cytokines predominate [39,42]. The second immunological phase of pregnancy is accompanied by rapid fetal growth. The mother, placenta and fetus maintain a harmonic relationship governed by an anti-inflammatory state with the prevalence of Th2-type cytokines [14]. In the last phase, the fetus has completed its development and childbirth is achieved through a renewed state of inflammation favored by Th1-type cytokines and an influx of immune cells into the myometrium. [43,44]. This pro-inflammatory milieu promotes the contraction of the uterus, leading to the expulsion of the fetus and its placenta. Thus, pregnancy switches between pro-inflammatory and anti-inflammatory states [31]. The participation of specific cellular and molecular elements defines each immunological stage. Several regulatory mechanisms control the balance between pro-inflammatory and anti-inflammatory reactions at the maternofetal interface. Epigenetic regulation of DNA methylation, imprinting and microRNAs (miRNAs) all play a pivotal role in these processes. MicroRNAs are involved in placental development and changes in their expression are related with pregnancy complications [9,45,46,47].

The balance and correlation between Th1 and Th2-type responses and their regulatory mechanisms at the maternofetal interface as well as their mediators should be further studied to provide a deeper understanding of the processes orchestrating the immune responses during pregnancy. As a result, new approaches to prevent implantation failure, recurrent abortions, premature birth, and other pregnancy complications related to immunological processes may arise [48].

Cytokines are key molecular messengers able to trigger, brake and balance the immune responses that take place at the maternofetal compartment. Following the recent description of the IL-36 cytokines, their expression, function in inflammatory responses, and relationship with the development of pathological conditions have been investigated. These cytokines gained attention due to their involvement in skin inflammatory processes caused by psoriasis [49]. In the following sections, we will describe the biological processes activated by IL-36 cytokines, their contribution to inflammatory conditions and, finally, focus on the emerging roles of these cytokines in normal and pathological pregnancies. Although information regarding IL-36 cytokines in reproductive processes still is incipient, studies on different inflammatory conditions and immune responses illustrate how these cytokines may participate in regulating human pregnancy.

## 2. The IL-1 Superfamily Members and Their Function in Reproduction

In the early 1980s, the prototype of the IL-1 cytokine superfamily was described. Previous to a detailed molecular characterization, many IL-1 functions were described under other names such as endogenous mediator of leukocytes, hemopoyetin 1, endogenous pyrogen, cataboline and osteoclast activating factor [50,51]. Later on, the IL-1 gene was characterized as a segment of approximately 450 kb on chromosome 2q14-q21. This region encodes also other cytokines that share a conserved gene structure, resulting in proteins with a high degree of similarity on their secondary and tertiary structures [52,53]. Due to these conserved characteristics, they are suggested to have arisen by duplications of a common ancestral gene [54,55]. Eleven members are currently known within this superfamily, namely IL-1α, IL-1β, IL-1Ra, IL-18, IL-33, IL-36α, IL-36β, IL-36γ, IL-36Ra, IL-37 and IL-38. These cytokines signal through their respective receptors, members of the IL-1 receptor family [56]. Based on precursor protein size, type of receptor used and performed activities, cytokines from the IL-1 superfamily are further divided into IL-1, IL-18, and IL-36 subfamilies (see Figure 1).

### 2.1. The IL-1 Subfamily

IL-1α, IL-1β, IL-1Ra, and IL-33 compose the IL-1 subfamily. The first two members have mainly pro-inflammatory Th17 and to a lesser extent Th1 activity when they bind to the receptor complex formed by IL-1R1 and IL-1RAcP (IL-1 receptor accessory protein), whereas IL-1Ra is a cytokine antagonist and IL-33 possesses Th2 properties [57]. IL-1α requires the processing of its precursor to generate the active form, which is performed by the membrane protease calpain or neutrophil-derived proteases [58]. This cytokine functions as an alarmin [59,60]. Its release generates an immediate response promoting the induction of several cytokines and chemokines that mediate so-called sterile inflammation [61]. Likewise, the 31 kDa IL-1β precursor is subjected to intracellular proteolytic processing performed by caspase-1, so that the active fraction of 18 kDa can be secreted [62]. Processing of the 30 kDa precursor fraction of IL-33 is also performed by neutrophil proteases to generate the 18 kDa active form [63]. IL-33 is responsible for a Th2-type inflammatory activity in infection and tissue repair [64]. IL-1Ra is the only molecule in the family that does not require cleavage and release of signal peptide for its secretion [65]. As an antagonist, binding of IL-1Ra to IL-1R1 prevents its coupling with IL-1RAcP and further signaling. Furthermore, the important regulatory activity of IL-1Ra in pregnancy was demonstrated in mice deficient for this molecule, which have fewer offspring exhibiting growth retardation. In addition, the prevailing pro-inflammatory condition in these animals leads to the development of chronic inflammatory polyarthropathy [66].

IL-1α and IL-1β have been associated with a post-coitus acute intrauterine inflammatory response in mice. The expression of IL-1β is also detected from day 3 until 9 of pregnancy with a peak between days 4-5, when the embryo implants and decidualization starts, indicating its relation to these processes [67]. In humans, although IL-1α, IL-1β and IL-1Ra show different concentrations in the cervicovaginal fluid, they remain relatively constant throughout pregnancy, followed by a rapid decline of IL-1Ra some days before delivery. These changes in the inflammatory balance of gestational tissues occur prior to the onset of labor [68]. IL-1β is associated with increased inflammation-like processes during delivery; its levels reach maximal values during labor and fall back in the postpartum period [69].

IL-33 is present in mouse ovaries along the estrous cycle with highest expression at diestrus and lowest at estrous [70]. IL-33 produced by macrophages acts as a regulator of trophoblast cell proliferation and placental growth [71]. Circulating and placental IL-33 remains almost constant throughout the entire human pregnancy, both in healthy and preeclamptic women. In contrast, there is an increase in the soluble fraction of IL-33 receptor (sST2) during the last trimester, with levels even higher in patients with preeclampsia [72].

### 2.2. The IL-18 Subfamily

The IL-18 subfamily consists of two members, IL-18 and IL-37. The IL-18 precursor is about 24 kDa and requires caspase-1 processing to generate the active 17.2 kDa form [73,74]. IL-18 is secreted by various cells such as macrophages, endothelial cells, keratinocytes, and intestinal epithelium, and DCs, among others. The IL-18 receptor complex formed by the IL-18Rα chain and the IL-18Rβ co-receptor is similar to that of other members of the IL-1 family. Receptor heterodimerization allows the Toll/IL-1 receptor (TIR) domains to generate a cascade of sequential recruitment. It begins with myeloid differentiation primary response 88 (MyD88) activation followed by IL-1 receptor associated kinase (IRAK) and TNF receptor-associated factor 6 (TRAF-6), leading to the degradation of IkB kinase complex (IKK) and release of nuclear factor kappa-light-chain-enhancer of activated B cells (NF-kB), which in turn favors the expression of pro-inflammatory mediators [65]. IL-18 has been described as a paracrine regulator of endometrial function. Elevated levels of IL-18 have been associated with implantation failure [75,76]. Human decidual and glandular cells express IL-18 whereas trophoblast cells express IL-18R, indicating its role in maternofetal communication. Moreover, IL-18 stimulates the cytotoxic capacity of uterine and peripheral blood NK cells [76,77]. Contrary to IL-33, during the mouse reproductive cycle, expression of IL-18 is higher at estrous than at diestrus. In addition, estradiol and progesterone negatively regulate this cytokine [78]. During normal human pregnancy, IL-18 expression is relatively high in the first and second trimester. At the labor stage, its levels are even more elevated. Furthermore, during several pregnancy disorders IL-18 concentrations are exacerbated, suggesting the participation of this cytokine in pathogenic processes [79].

IL-37 is the antagonist member of the IL-18 subfamily. An active form of 22 kDa is generated by cleavage through caspase-1 and binds to IL-18Rα, leading to inhibition of this receptor [80]. The participation of IL-37 in pregnancy related processes remains unknown.

### 2.3. The IL-36 Subfamily

Around a decade ago, new members of the IL-1 family were discovered based on their homology at both gene and protein levels. Suggested names were initially derived from the order of their discovery. They were designated from IL-1F1 to IL-1F10 to unify the nomenclature and to ease their identification since several groups had described variants of the same molecules [54]. Upon the description of their particular biological functions, the assignment of unambiguous names for each molecule was proposed. Currently, IL-36α (IL-1F6), IL-36β (IL-1F8), IL-36γ (IL-1F9), IL-36Ra (IL-1F5), and IL-38 (IL-1F10) are collectively known as IL-36 subfamily cytokines [54,55].

## 3. Biology of the IL-36 Subfamily 

The IL-36 subfamily comprises a group of cytokines with similar characteristics to IL-1. IL-36α, IL-36β and IL-36γ are agonist ligands with pro-inflammatory activity. They promote the induction of various inflammatory mediators including cytokines, chemokines, growth factors, and antimicrobial peptides [49,81]. All of them use the same receptor, IL-36R, which dimerizes with IL-1RAcP to activate intracellular signaling cascades. This pathway, described in a later section, culminates with the expression of inflammatory cytokines driven by AP-1 (activator protein 1) and NF-kB transcription factors (See Figure 2). In contrast, IL-36Ra exerts antagonistic effects by inhibiting receptor dimerization and the activation of its signaling mediators. To date, IL-38 has not been characterized extensively. Evidence indicates this cytokine functions as a negative regulator similarly to IL-36Ra [80,82,83]. Each member of this subfamily has particular actions, whose effects mediate inflammatory processes.

All members of the IL-36 subfamily have been identified in both humans and mice. They share a considerable degree of sequence homology, respectively 91%, 54%, 62% and 56% for IL-36Ra, IL-36α, IL-36β and IL-36γ. In addition, the location and general organization of the IL-1 and IL-36 loci are similar between humans and mice, highlighting the importance of these cytokines across species [52].

### 3.1. Processing and Secretion of IL-36 (α, β, γ) and IL-36Ra.

Initial analyses of human epithelial and Jurkat T cell cultures revealed a high concentration of IL-36Ra in supernatants and, to a lesser extent, in the cytoplasm, indicating the secretion of this cytokine [84]. While cleavage of IL-1β and IL-18 by caspase-1 promotes their activation and secretion [85,86], the 17–20 kDa protein encoded for IL-36Ra lacks N-glycosylation at Asn91, a caspase cutting site and conventional leading peptide sequence [87], which suggests an unconventional secretion pathway compared to other members of the IL-1 family.

Similarly to IL-36Ra, IL-36α, IL-36β and IL-36γ do not present a characteristic cutting site for caspase-1, and thus far, no signal peptide or pro-domain has been found in any of them. Therefore, their secretory processes are not yet known [88]. However, significant amounts of these cytokines have also been found in cell culture supernatants suggesting an alternative mechanism for their secretion [89,90].

Reports elucidating the molecular processing and secretory mechanisms employed by IL-36 cytokines have begun to emerge. Transfection of bone marrow derived macrophages (BMDMs) to constitutively produce IL-36α demonstrated that it does not derive from the endoplasmic reticulum and accumulated intracellularly. Only combined LPS/ATP (adenosine triphosphate) stimuli trigger the secretion of IL-36α, without any apparent proteolytic process. The addition of CP-456773, an inhibitor of ATP activity, completely abolished IL-36α secretion. Moreover, treatment with nigericin, an inducer of the ionophore P2X7 receptor (a two-transmembrane ionotropic receptor), increases its release. Treatment with cycloheximide, however, does not affect either IL-36α expression levels or its secretion. When using a caspase-1 inhibitor (Ac-YVAD-CMK), the production and externalization of IL-36α is not affected. K+ excess, known to block the activation of the inflammasome component NALP3, markedly decreased IL-36α secretion. This comprehensive set of experiments demonstrated that a K+ sensitive inflammatory complex is required for IL-36α secretion by LPS/ATP-treated BMDMs, independently of caspase-1 processing. It also showed that IL-36α secretion occurs via a non-selective mechanism that may be related with plasma membrane rupture, TLR-dependent signaling, priming independent of TLR as well as ATP-dependent P2X7 activation [91]. 

Other mechanisms for processing of IL-36 cytokines require caspase activation. For instance, blocking of caspase-3 activity interfered with the release but not with the expression of IL-36γ, which accumulates intracellularly. On the other hand, inhibition of caspase-1 suppresses both the expression and secretion of IL-36γ [92].

In order to acquire its antagonistic activity, IL-36Ra needs proteolytic processing. A methionine located at the N-terminus is removed leaving a valine at this position, which facilitates IL-36Ra interaction with IL-36R leading to its inhibition. Similar processes mediate the activation of IL‑1Ra and its binding to IL-1R. Due to this observation, it has been investigated whether a similar mechanism to IL-36α, IL-36β, and IL-36γ may be necessary to confer activity. Previous studies reported that large quantities of the full-length ligands were necessary to generate an inflammatory response [86,93]. This low activity could be attributed to the lack of proteolytic processing that generates the active forms. In this way, the truncation of 9 amino acids in the N-terminal portion from the A-X-Asp motif, which is conserved among the members of the IL-1 family, was analyzed for each cytokine (See Figure 1). Excision at the K6 amino acid for IL-36α, at R5 for IL-36β, and at S18 for IL-36γ increased their affinity to IL-36R respectively in ~36,000-, ~18,000 and ~980-fold. Accordingly, truncated ligands display 1000- to 10,000-fold increase in their activities compared to the full-length ones [84].

Proteases released by activated neutrophils differentially process and activate IL-36 cytokines (α, β, γ). Cathepsin G, for instance, selectively promotes IL-36β cleavage, whereas elastases and, to a lesser extent, proteinase-3 process IL-36γ. Both elastase and cathepsin G are responsible for IL-36α activation. The cleavage sites for cathepsin G and elastase on IL-36α are Lys3 and Ala4, respectively. On IL-36β, cathepsin G cleaves at Arg5. Elastase and proteinase-3 cut IL-36γ at Val15 [94]. Finally, neutrophil-derived elastase is the protease responsible for cleavage at Val2 to obtain the active antagonistic form of IL-36Ra [95]. As typical for some IL-36 family members, IL-38 lacks a signal peptide and caspase-1 consensus cleavage site [82,96]. Based on a predicted cleavage site for IL- 38 in mice, a consensus cleaving site has been suggested that predicts the removal of the first two amino acids to generate a processed IL-38 protein [83,97]. However, the natural N-terminus of IL-38 is still unclear.

### 3.2. The IL-36 Receptor Complex

The first IL-1 receptor (IL-1R) was described in 1988 [98]. Since its discovery, a great number of publications have identified and characterized various receptors which share similar structures leading to identification of at least ten members of the IL-1 receptor family [99]. In this family, an additional protein related to the biological activity of IL-1 was described. IL-1RAcP belongs to the super family of immunoglobulins due to the presence of three respective domains on its extracellular fraction. To generate a functional receptor for IL-1α and IL-1β, IL-1RAcP forms a heterodimeric complex with IL-1R [100,101].

IL-36R was described in 1996 as an orphan receptor belonging to the IL-1 receptor family and denominated IL-1 receptor related protein two (IL-1R-rp2) [102]. IL-36R consists of an extracellular, a transmembrane, and an intracellular portion. Within the extracellular portion of IL-1 receptor family members, three immunoglobulin domains are equally conserved. The intracellular portion contains the Toll-like domain that is involved in the type of signaling generated when IL-36 agonists (α, β or γ) bind to IL-36R [102,103].

Jurkat T cells transfected with IL-36R respond to stimulation with IL-36γ. Due to the homology of IL-36R with IL-1R, the authors suggested that IL-36R needs another auxiliary receptor to function [86]. Indeed, IL-1RAcP has been demonstrated to be an essential co-receptor to trigger the signals of IL-36 cytokines through IL-36R [93]. The truncated fraction of IL-36 (α, β, γ) binds to the extracellular portion of IL-36R with high affinity (*Kd = 10^−5^ s^−1^* M) and favors the recruitment of IL-1RAcP but does not associate directly to it. The dimerization of IL-36R:IL-1RAcP promotes the phosphorylation of TIR domains present on each subunit, which induce activation of intracellular signaling pathways (See Figure 2) [84]. IL-36 (α, β, γ) and IL-36Ra have a residue in their C-terminal portion within the loops β4/5 and β11/12 of their secondary structure which confers them their agonist or antagonist characteristic. Asp150 for IL-36α, Asn148 for IL-36β, Ala162 for IL-36γ, and Asp148 for IL-36Ra form hydrogen interactions with IL-36R that help maintain the three-dimensional orientation of IL-1RAcP in the IL-36 quaternary complex (IL-36R:IL-1RAcP). This process allows the interaction of IL-36R with Ser185 of IL-1RAcP. In contrast, in the longer β11/12 loop of IL-36Ra, the Asp148 residue functions as a steric obstacle that prevents the union of IL-1RAcP with IL-36R [104]. These molecular differences in the subfamily of IL-36 members confer individual activating or inhibiting properties to each member in the context of inflammatory responses. Depending on the cell type, functional context, and interaction with other mediators, IL-36 cytokines may have specific effects and regulatory mechanisms despite using the same receptor.

### 3.3. IL-36-Induced Signaling Pathways 

The initial characterization of members of the IL-1 receptor family (IL-1Rs) suggested the existence of additional ligands that could bind to orphaned receptors and function as independent signaling systems. By 2001, it had been identified that IL-36γ mediates the activation of NF-kB through IL-36R in Jurkat T cells [86]. Following, IL-36α, IL-36β and IL-36γ have been demonstrated to activate NF-kB in an IL-36R-dependent way without the participation of other members of the IL‑1R family. MAPKs, JNKs, and ERK1/2 mediate the activation of NF-kB by IL-36 cytokines. These pathways show a peak of activation 15 min after IL-36 treatment and lead to a rapid increase in the phosphorylation of IkB-α [93,105]. IL-36β specifically activates the phosphorylation of p38 MAPK in BMDCs [106]. In bronchial epithelial cells, IL-36γ activates MAPKs, JNK, ERK and p38 MAPK as well as NF-kB and CREB transcription factors [107]. Overall, the expression of IL-36α, IL-36β and IL-36γ positively correlates with the phosphorylation of p38 MAPK and p65 or RelA, a member of NF-κB transcription factor family [108]. Furthermore, stimulation of HT-29 (human colon cancer cell line) and WiDr (colon adenocarcinoma line) cells with IL-36α and IL-36γ activate a signaling cascade through MyD88 adaptor protein complex (MyD88, TRAF6, IRAK1, and TAK1), phosphorylation of MAPKs and activation of NF-kB and AP-1 [109]. Much remains to be clarified about the signaling mechanisms induced by IL-36 cytokines in order to understand their inflammatory and other biological processes.

### 3.4. IL-36Ra and IL-38: The Negative Regulators of IL-36R Activity

IL-36Ra lacks the ability to induce the expression of cytokines or to block the activity of IL-1α or IL-1β [85]. Instead, IL-36Ra antagonizes IL-36γ-mediated activation of NF-kB through IL-36R, as a result of the absence of a loop between the 4-5 folded beta sheets. This region is a site where amino acid residues strongly interact with their respective receptor and favor the binding with IL-1RAcP [86]. IL-36Ra also inhibits the effects of IL-36α in vivo in mouse models that overexpress this agonist [110]. In addition, IL-36Ra acts as a selective inhibitor of IL-36α, IL-36β and IL-36γ in BMDCs. This inhibitory process occurs in a dose-dependent manner but requires a molar excess of 100 to 1000 times [106]. After demonstrating that IL-36Ra needs processing to remove a methionine from its N-terminal and generate the active fraction, it has been found that the mechanism of action of IL-36Ra is very similar to that of IL-1Ra. Through the currently accepted mechanism, IL-36Ra binds to the extracellular domain of IL-36R impeding the interaction of this receptor with IL-36α, IL-36β, and IL-36γ. Consequently, the recruitment of IL-1RacP to generate the IL-36R:IL-1RAcP complex and the corresponding signals are arrested (see Figure 2) [84].

IL-38 inhibits the production of IL-17 and IL-22 by T cells challenged with *Candida albicans* components. IL-38 also suppresses IL-8 in peripheral blood mononuclear cells (PBMCs) stimulated with IL-36γ, in the same way as IL-36Ra does [83]. In addition, in vitro administration of IL-38 counteracts the biological effects induced by IL-36γ in human keratinocytes and endothelial cells. In vivo, this cytokine attenuates the severity of the imiquimod (IMQ)- induced psoriasis in a mouse model [111]. There is also evidence supporting the anti-inflammatory effects of IL-38 in macrophages leading to reduced Th17 expression. Based on this property IL-38 may be used for targeting numerous inflammatory pathologies [112].

Currently, IL-36Ra and IL-38 are the only known negative regulators of the IL-36 axis, leading to speculation on the existence of additional mediators and molecular mechanisms acting on the fine tuning of this system. 

## 4. IL-36(α, β, γ) Cytokines as Amplifiers of the Inflammatory Response

The expression of IL-36 cytokines has been found in many cellular types. Despite of this, the pathophysiology of the inflammatory response triggered by this group of cytokines seems to have a common feature, an inflammatory triad restricted to epithelial cells, neutrophils and DCs. In turn, these cells secrete, activate and amplify the inflammatory signal of the IL-36 axis from an innate to an adaptive immune response. This process involves distinct cell types depending on the tissue and respective pathology (see Figure 3).

### 4.1. Skin Immunopathophysiology as a Model to Understand The IL-36-Trigered Inflammatory Response

Little is known about the molecular and cellular mechanisms controlling the initial stages of IL-36-triggered inflammation. Psoriasis represents a valuable model to understand the events related to IL-36 activity and function [113]. A complex cross-talk between keratinocytes, neutrophils, skin-resident DCs, infiltrating macrophages, T cells, and other immune cells is associated with uncontrolled keratinocyte proliferation and dedifferentiation as well as angiogenesis [114,115]. In subjects who are genetically predisposed, the activation of TLR9 on DCs by drugs, pathogen-associated molecular patterns (PAMPs) or other environmental factors promote the release of IL-36 (α, β, γ), IL-12 and IL-23. These cytokines drive polarization into effector T cells secreting TNF-α, IFN-γ, IL-17, IL-22 and other inflammatory mediators. Cytokines released under the influence of IL-36 (α, β, γ) synergize and stimulate epithelial cells to produce a variety of growth factors and inflammatory mediators fueling and amplifying a vicious cycle in this skin pathology. DCs are crucial for the induction of IMQ-driven psoriasis. The absence of these cells protects mice from ear swelling, neutrophil infiltration, T cell expansion, and the development of IL-17–producing cells. Stimulation of DCs with IMQ or IL-36β induces the expression of IL-23, IL-36α, and IL-36γ [116].

Keratinocytes and other cells are major sources of IL-36 cytokines, particularly during inflammatory processes. IL-36γ appears to be the most characteristic marker in human psoriatic skin lesions. Expression of IL-36 (α, β, γ) in these lesions is higher compared with other skin diseases [117]. IL-36 enhances the production of psoriasin and LL37, both overexpressed in several skin diseases. LL37 is a mature cathelicidin peptide derived from human cathelicidin (hCAP18) through enzymatic cleavage by kallikreins in the epidermis. This peptide kills microbes and exerts alarmin activity, serving as an innate immune effector that recruits and activates immune cells as well as induces pro-inflammatory cytokines and chemokines [118]. In the skin, LL37 allows plasmacytoid DCs to recognize self-DNA through TLR9 and stimulates keratinocytes to induce more TLR9. In this milieu, LL37 induces IL-36R and IL-1RAcP as well as IL-36γ, IL-36β, and other IL-1 family genes. LL37 induces IL-36γ in both undifferentiated and differentiated keratinocytes favoring the induction of chemokines by itself and in combination with IL-36γ. IL-36γ also induces LL37 in a positive feedback loop in lesional psoriatic skin. Interestingly, when IL-36γ or IL-36R are blocked, the LL37-dependent production of IL-8, CXCL1, CXCL10, and CCL20 in human keratinocytes is suppressed, showing that the alarming functions of LL37 in the human epidermis are enhanced by IL-36γ. Induction of IL-36γ by LL37 contributes to the mechanism through which the innate response is initiated in skin inflammation and exacerbated psoriatic lesions via chemokines, cytokines and growth factors, mediators that induce the recruitment and activation of DCs, neutrophils, macrophages, T cells and NK cells [118].

Treatment of human keratinocytes with IL-36 (α, β, γ) induces the expression of chemotactic agents such as CXCL1, CXCL8, CXCL10, CCL2, CCL3, CCL5, and CCL20. When IL-36α is injected intradermally into the mouse skin, it enhances chemokine and growth factor expression, leukocyte infiltration, and acanthosis. Blood monocytes, myeloid dendritic cells, and monocyte-derived dendritic cells (MDDCs) express IL-36R and respond to IL-36 (α, β, γ) [119]. In addition, IL-1α and TNF induce IL-36 (α, β, γ) and IL-36Ra expression in human keratinocytes from inflamed epithelia. These cytokines stimulate the production of antimicrobial peptides and matrix metalloproteinases in reconstituted human epidermis. IL-36β increases mRNA expression of human beta-defensins (HBD) and cathelcidin, underlining the important role of IL-36 cytokines in psoriatic inflammation [120]. Collectively, IL-36 cytokines induce, maintain and expand the innate immune response by stimulating the release of cytokines, chemokines, and growth factors in association with the recruitment of immune cells to lesioned skin.

Psoriasis pathophysiology involves both innate and adaptive immune responses to exacerbate skin inflammation. Keratinocytes, DCs, neutrophils, macrophages and CD4+ T cells express IL-36R and respond to IL-36 (α, β, γ) [106]. In DCs, IL-36γ shows a constitutive expression pattern while IL-36α is inducible [121]. When DCs are stimulated with IL-36 cytokines, they show a strong induction of IL-6, IL-12p40, CXCL1, CCL1, IL-12p35, IL-1β, IL-23p19, granulocyte-macrophage colony-stimulating factor (GM-CSF), IL-10, CXCL10, tumor necrosis factor alpha (TNF-α), cyclooxygenase-2 (COX-2), and nitric oxide synthase 2 (NOS2). Interestingly, IL-36 cytokines can trigger DC maturation by increased expression of MHC-II, CD80, and CD86. To complete the immune response network, CD4+ T cells also express IL-36R and respond to IL-36 stimulation. Th1 cells express higher levels of IL-36R than Th2 and Th17 cells. Stimulation of CD4+ T cells with IL-36 cytokines potently induces the production of IFN-α, IL-4, and to a lesser degree IL-17. IL-36 cytokines modulate splenocyte proliferation and the production of several chemokines, cytokines, and growth factors such as granulocyte-colony stimulating factor (G-CSF), GM-CSF, and the vascular endothelial growth factor (VEGF). This may be due to a constitutive expression of IL-36β and IL-36γ. Specifically, IL-36β acts as an adjuvant to stimulate the Th1 immune response through a robust induction of IFN-γ in mice immunized with a mix of mBSA/IL-36β. Thus, it is plausible to assume that IL-36 (α, β, γ) acts as a bridge in the activation of innate and adaptive immune responses, fostering IL-1β, IL-6, TNF-α, and IL-23p19. These cytokines have been shown to be involved in the generation of Th17 immune response, which plays a relevant role in the pathophysiology of skin inflammation [106,122].

When treated with IL-36 (α, β, γ), monocytes express IL-1α, IL-1β and IL-6 whereas myeloid DCs upregulate their surface expression of costimulatory CD83, CD86 and HLA-DR molecules. MDDCs stimulated with IL-36α enhance allogeneic CD4+ T cell proliferation. IL-36 may influence T cell function in skin via its effects on APCs and their expression of Th17-inducing IL-1β and IL-6. These data indicate that IL-36 (α, β, γ) actively amplifies inflammation via activation of epithelial cells, APCs and T cells [119].

Psoriasis appears to involve a cytokine network centered on IL-17, IL-22, IL-23 and TNF. All of these are elevated in lesioned skin, where they cause epidermal hyperplasia by down-regulation of genes involved in terminal keratinocyte differentiation and cytokine induction toward Th1/Th17 immune responses. IL-17, IL-22, and IL-23 are involved in skin inflammation and highly induced in mice that overexpress IL-36α or in those treated with 12-O-tetradecanoylphorbol-13-acetate to induce psoriatic lesions. Together, they establish a self-amplifying cytokine-expression loop. IL-36α transgenically expressed in keratinocytes acts in an autocrine and paracrine fashion on skin DCs and macrophages to enhance the synthesis of cytokines, chemokines, and antimicrobial peptides [94]. The inflammatory response observed in this model is a typical feature of the psoriatic phenotype. Genetically manipulated animal models lacking IL-36R or IL-1RAcP do not exhibit a psoriatic skin phenotype [85]. Consistently, IL-36Ra^–/–^ mice show exacerbated disease, supporting the regulatory activity of this antagonist in vivo [88]. Thus, a disbalance between IL-36 (α, β, γ) and IL-36Ra contributes to the pathophysiology of skin inflammation [85,94]. The associated cellular and molecular mechanisms may be related to those in other pathologies where the IL-36 cytokine system is altered, including in pregnancy complications, as discussed in the next sections.

### 4.2. Inflammatory Pathologies Related with the IL-36 Dysregulation 

Due to the ability of IL-36 (α, β, γ) to induce skin inflammation, its involvement in other inflammatory conditions has begun to be examined. IL-36α has been associated with the development of renal pathologies, including glomerulonephritis, where it is overexpressed in distal convoluted tubules and in cortical collecting ducts. IL-36α correlates with the severity of the tubular damage in the B.6MRLcl chronic glomerulonephritis mouse model, streptozotocin-induced diabetes, and in models of lupus erythematosus such as BXSB, NZB/W F1 and MRL/lpr. In addition, the overexpression of IL-36α is associated with a marked infiltration of CD3+ mononuclear cells and the presence of nestin and α-SMA-expressing cells, indicators of renal interstitial fibrosis. Accordingly, increased levels of IL-36α have been detected in renal biopsy specimens and urine samples from patients with renal tubulointerstitial lesions (TILs). They correlate with renal function impairment [123].

Elevated expression of IL-36α in renal tubular epithelial cells was described in mice with unilateral ureteral obstruction. In kidneys of IL-36^-/-^ mice with unilateral ureteral obstruction, TILs are diminished in association with a marked reduction of NLRP3 inflammasome activation and infiltration of macrophages and T cells. In this model, IL-23 and IL-17A are deficient which may contribute to reduced TIL formation. In vitro, recombinant IL-36α facilitates NLRP3 inflammasome activation in renal tubular epithelial cells, macrophages, and dendritic cells as well as enhances DC–induced T cell proliferation and Th17 differentiation. These results indicate a synergistic interaction between the IL-36 and the IL-23/IL-17 axis in the amplification of inflammatory disorders [124].

The IL-36R axis has been related also to the development of inflammatory lung processes. In cultures of human bronchial epithelial (NHBE) cells, the expression of IL-36 (α, β, γ) was found to be induced by cytokines (IL-1β, IL-17, IFN-γ), TLR ligands (dsRNA, LPS, flagellin, FLS-1), and other factors. TNF-IL-17 combination or dsRNA seem to be the strongest inducers of IL-36γ expression. Furthermore, normal human lung fibroblasts (NHLF) stimulated with IL-36γ secrete high levels of IL-8, CXCL3, CCL2, CCL5, CXCL10, G-CSF, GM-CSF, CCL20, IL-1β and IL-6. This cytokine profile suggests that IL-36γ is involved in neutrophil recruitment and infiltration leading to increasing inflammation in the airways [107]. In a mouse model of allergic lung inflammation, IL-36α and IL-36γ are predominantly elevated in airway epithelial cells. In addition, intratracheal administration of IL-36γ causes epithelial cell hypertrophy and increases neutrophil infiltration. IL-36α also favors the expression of TNF, IL-1α and IL-1β. The administration of these cytokines to experimental animals induce the expression of CXCL1 and CXCL2, which are major chemoattractants for neutrophils. Both IL-36α and IL-36γ increase the activation of NF-kB. IL-36γ downregulates several nuclear factors such as AR, hypoxia-inducible factor (HIF), MEF2, NF1, NkX2-5, ISRE and MyoD which promotes proliferation of CD4+ T cells and CD40 expression in spleen CD11c+ cells. These data suggest a unique mechanism by which IL-36 cytokines initiate the inflammatory response in the respiratory tract [125]. Furthermore, the stimulation of bronchial, endothelial and alveolar epithelial cells with IL-36 (α, β, γ) induces the production of inflammatory mediators and the recruitment of neutrophils and lymphocytes. As a result, fibrotic pathways related to WNT5A and BMP2 get activated leading to pulmonary fibrosis [126].

IL-36α has also been involved in the prognosis of hepatocellular carcinoma, where its expression correlates negatively with tumor size, degree of differentiation, and level of vascular invasion. Correspondingly, high levels of IL-36α are positively correlated to the overall survival of patients and with decreased tumor growth promoted by the recruitment of CD4+ and CD8+ T cells [127]. In addition, IL-36γ has an important role for anti-tumor immune responses by transforming the tumor microenvironment and promoting the differentiation of type 1 effector lymphocytes. Elevated expression of IL-36γ correlates with an increased number of tumor-infiltrating lymphocytes (including CD8+, NK, and γδ T cells) resulting in strong anti-tumor activity. IL-36γ enhances the adaptive tumor antigen-specific CD8+ T cell immune response. In a murine in vitro model, this anti-tumor effect depends on IL-36R expressed in CD8+, NK and γδ T cells, whose signaling directly promotes proliferation and IFN-γ production. Interestingly, tumor cells expressing IL-36γ function as an effective tumor vaccine suggesting that this cytokine has implications for anti-tumor immune therapy. Furthermore, IL-36γ promotes IFN-γ production in human CD8+ T cells and its expression inversely correlates with melanoma and lung cancer progression [128].

The IL-36 axis has been implicated in experimental colitis and human inflammatory bowel disease. IL-36γ is expressed in both conditions and the gut microbiota appears to be its inducer. Germ-free mice fail to produce IL-36γ in response to dextran sodium sulfate (DSS)-induced injury. IL-36R knockout mice exhibit defective recovery and impaired closure of colonic mucosal wounds, compromised neutrophil accumulation in the wound bed and decreased neutrophil-derived IL-22 expression. IL-36R contributes to DSS-induced IL-22 production, a barrier-protective cytokine that stimulates epithelial proliferation and restitution, induces secretion of antimicrobial peptides and protects from intestinal inflammation, suggesting that the IL-36 axis has an important role in the resolution of intestinal mucosal wounds [129,130]. In addition, IL-36α and IL-36γ over-expression and IL-38 reduction have been found in the inflamed colon of a minor subpopulation of patients with Crohn’s disease [131].

The participation of IL-36 (α, β, γ) in infection by *Mycobacterium (M). bovis* BCG and *M. tuberculosis* H37rv has been studied. Although IL-36γ is highly expressed, the CD4+ T cell antigen-specific response is poor and the production of IL-6, TNF-α, IL-12p40, IL-4, IL- 13 and IL-17 is low. In this context, the Th1 immune response against BCG and *M. tuberculosis* occurs in an IL-36-independent way [122,132].

IL-36Ra decreases IL-6-driven inflammation as well as LPS- and IL-1β-induced MAPKs activity in hippocampal glial cells in vitro and in vivo. These responses are partially dependent on IL-4 induction and IL-36Ra activity in glia cells. Interestingly, the inhibitory action of IL-36Ra is related to its interaction with the SIGIRR receptor (IL-1 receptor accessory protein-like and single receptor IL-1 receptor-related molecule). IL-4 is not detectable in cells from SIGIRR-deficient mice treated with IL-36Ra [133].

IL- 36β and IL-36R are basally expressed in particular cells from healthy patients and become significantly elevated in patients with rheumatoid arthritis or osteoarthritis [134]. IL-36 (α, β, γ) and IL-36R have been found in inflamed joints of mice from collagen-induced arthritis, antigen-induced arthritis, TNF-induced arthritis and K/BxN serum transfer-induced arthritis. Blockade of IL-36R does not have apparent effects nor do IL-36R^-/-^ mice show improvements in arthritis [135,136]. These data demonstrate that IL-36R signaling is not essential to the course and severity of this disease but does not rule out its contribution. Its effects may be compensated by other cytokine systems through a potential redundancy of the IL-36 system.

Although the involvement of IL-36 in several physiological and pathological situations has been investigated the global and specific knowledge on IL-36 is still limited. It may be expected that a more detailed understanding may lead to novel clinical implications and facilitate the development of strategies for treatment of inflammatory disorders.

## 5. IL-36 Cytokines in Female Reproductive Tissues and Pregnancy

As described above, IL-36α, IL-36β, IL-36γ, IL-36Ra and their receptor IL-36R are expressed by several immune and non-immune cell types. Epithelial cells, granulocytes, DCs, macrophages and T cells are potent producers of and responders to these cytokines [81,113]. Their role in innate and adaptive immune responses associated with the enhancement of inflammation in skin, kidneys, joints, brain, and lungs has begun to be disclosed. [137]. However, there is little knowledge regarding the role of IL-36 cytokines in the reproductive organs in pregnant and non-pregnant females.

The expression of members of the IL-36 cytokine system has been demonstrated in tissues and organs of the male and female reproductive systems. For example, IL-36R has been found in testicles and epididymis [102], while IL-36Ra, IL-36α and IL-36β are found in the prostate [88]. Expression of IL-36 cytokines has also been described in mouse and human uteri [89]. All IL-36 agonists are also expressed in the human placenta [88] and in the trophoblastic cell lines JEG-3, BeWo, SGHPL-5, AC1-M59 and JAR [85].

IL-36 cytokines are differentially expressed and regulated in several cells such as keratinocytes, fibroblasts, enterocytes, monocytes and osteoclasts stimulated with inflammatory cytokines or TLR ligands including poly I:C and LPS [131]. Poly I:C seems to be the strongest IL-36 inducer and particularly low doses induce expression and release of IL-36γ in a dose- and time-dependent manner in keratinocytes [138]. IL-36γ and IL-36R are present in epithelial cells of the human female reproductive tract (FRT) and are differentially induced by poly I:C treatment in vaginal and endocervical epithelial cells. This report documented the participation of IL-36γ in anti-viral defense and amplification of immune responses in epithelial cells from the lower FRT mucosa [139]. IL-36γ can be induced by HSV-2 infection in a similar way as by poly I:C treatment in a three-dimensional human vaginal epithelial cell model. Pre-treatment with IL-36γ prior to mouse intravaginal HSV-2 challenge significantly limits vaginal virus replication, delays disease onset, decreases disease severity and increases survival of infected mice. IL-36γ induces a transient production of immune mediators and cellular recruitment in the vaginal microenvironment increasing resistance to HSV-2 infection and disease. These studies provide evidence supporting the role of IL-36γ in host defense of the lower FRT [140]. In contrast, there is almost no evidence about IL-36 in the upper FRT.

Previously, we have reported the uterine expression profile of all members of the IL-36 subfamily during estrous cycle and pregnancy in mice. During the estrous cycle, uterine IL-36 (α, β, γ), IL-36Ra and IL-36R present a cyclic expression pattern. The lowest mRNA and protein levels have been observed in diestrus and the highest at estrus, suggesting their expression is under hormonal control (see Figure 4). IL-36 members are localized in the uterine luminal and glandular epithelia and in a few cells of the stroma at estrus. From day 4.5 to 10.5 of pregnancy, IL-36α and IL‑36β mRNA and protein tend to decrease and IL-36γ to increase, suggesting the potential activity of the IL-36 system in early pregnancy. IL-36R is consistently expressed during the same period. Interestingly, on day 16.5 of pregnancy, we found increasing pro-inflammatory IL-36(α, β, γ) and antagonistic IL-36Ra with a peak during labor, and a fallback in the postpartum period. These results indicate an involvement of the IL-36 system in processes that induce or accompany labor.

Deregulated expression of IL-36 cytokines could be associated with the development of pregnancy complications. Using a mouse model of pregnancy infected by *Listeria monocytogenes,* we described a differential and strong mRNA and protein expression of IL-36 (α, β, γ) as well as IL-36Ra and IL-36R in the uterus [141]. We also observed that high levels of IL-36 cytokines in infected animals enhances inflammatory activity, associated with a high ratio of fetal resorption [142]. The precise roles of IL-36 cytokines in bacterial infections remain to be elucidated.

A recent study reported the expression of IL-36 (α, β, γ) and IL-36Ra in the placenta of women with normal pregnancy and preeclampsia. Although the concentrations of IL-36 (α, β, γ) in serum do not change, IL-36Ra is expressed at lower levels in patients with preeclampsia who underwent emergency cesarean at 27–39 weeks of gestation compared with patients with normal pregnancy under elective cesarean at term (>38 weeks). Histologically, IL-36Ra is located around fetal blood vessels of the placental villi. Serum IL-36Ra is significantly higher in later stages of normal pregnancy compared to non-pregnant women. IL-38 seems to be also decreased in pre-eclamptic placentas [143].

Although inflammation during pregnancy is critical for proper implantation, decidualization and labor initiation, overlapping their physiological thresholds (see Figure 4, dotted red line), may be involved in pregnancy complications. Pathogen-Associated Molecular Patterns (PAMPs) and damage-associated molecular patterns (DAMPs) are able to induce the expression of pro-inflammatory cytokines, including members of IL-1 family, in placental tissues [144,145]. This information is in line with our findings showing that IL-36 cytokines can be induced by PAMPs.

## 6. Future Issues for IL-36 Cytokines in Pregnancy

The emergence of IL-36 cytokines as important mediators of inflammatory diseases in various tissues, including the placenta, has unveiled a range of potential novel mechanisms and targets for the study of inflammatory events related to pregnancy complications. The immediate reasoning may suggest that pro-inflammatory IL-36 (α, β, γ) overexpression has implications in the pathogenesis of placental pathologies due to their capacities of amplification of inflammatory responses. IL-36Ra and IL-38 are also relevant players in this scenario; reduction of their levels is associated with the development of inflammatory processes and contributes to dysregulated pro-inflammatory IL-36 (α, β, γ) activity [146,147,148,149].

Nevertheless, more details of the expression patterns, cellular sources, and functions of IL-36 cytokines during pregnancy in humans and experimental models need to be unveiled. A deeper understanding of the pathophysiology of immune and inflammatory disorders of the FRT and pregnancy, and particularly of placental pathologies may be expected in future. These data will aid in the establishment of theoretical and experimental approaches for potential diagnostic and therapeutic interventions.

Despite sharing the IL-36R, it is becoming evident that IL-36 cytokines can be individually regulated and can trigger distinct regulatory mechanisms in inflammatory processes. Identification of the transcriptional network engaged in the expression of these cytokines represents a pivotal step for further understanding their regulation. A better characterization of the downstream pathways driven by IL-36 cytokines will contribute to the development of strategies to regulate their activity in inflammatory conditions.

An approach to address mechanisms affecting IL-36 activity in pregnancy complications involves the study of proteases. Cathepsin G, proteinase-3, and elastase are able to process IL-36 family members into their active forms. Activation of neutrophils, a major source of these proteases, constitutes a feature of women with preeclampsia [150,151]. Enzymes released by placental neutrophils into the maternofetal interface have the potential to modify IL-36 activity directly or indirectly. Their subsequent effects comprise processes involved in the arteriopathy and endothelial damage associated with preeclampsia, for instance, up-regulation of cellular adhesion molecules on the endothelial surface, increased generation of angiogenic factors, and endothelial activation [114].

So far, the IL-36 cytokine system has been studied essentially in regard to its pro-inflammatory properties associated with disease development. The regulatory mechanisms controlled by IL-36 cytokines under physiologic conditions have been only superficially considered, as in the case of reproductive processes.

## 7. Concluding Remarks

The IL-36 subfamily constitutes a group of pleiotropic cytokines with multiple local and systemic effects. These molecules coordinate regulated pro-inflammatory processes involved in both innate and adaptive immune responses. Reviewed data highlight the spectrum of functions of IL-36 cytokines in defense mechanisms and inflammatory conditions in different tissues. They form part of the regulatory immune milieu at the maternofetal interface in healthy and complicated pregnancies, but their exact contribution is still unclear. The expanded knowledge about antagonist molecules and membrane decoy receptors, activated signaling pathways, involved microRNAs and epigenetic changes will provide a more elaborated scenario regarding the biology of IL-36 cytokines.

## Figures and Tables

**Figure 1 ijms-20-01649-f001:**
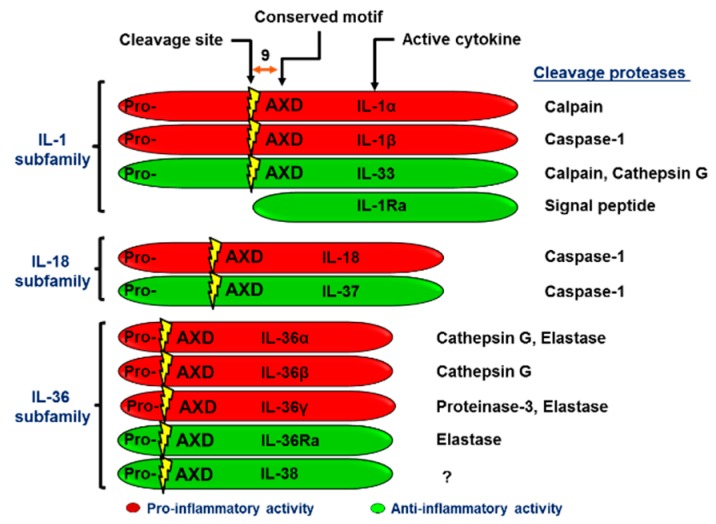
IL-1 superfamily. The IL-1 superfamily is comprised of 11 cytokines with pro-inflammatory or anti-inflammatory activity divided into IL-1, IL-18, and IL-36 subfamilies. All members, except IL-1Ra, are synthesized as long precursor proteins, which are proteolytically cleaved by the indicated enzymes to generate their active mature proteins.

**Figure 2 ijms-20-01649-f002:**
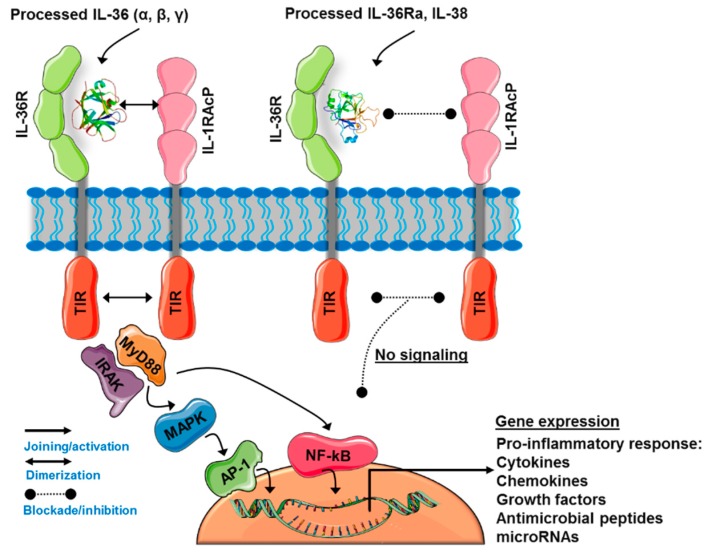
IL-36 signaling pathway. IL-36α, IL-36β, and IL-36γ bind to IL-36R leading to its dimerization with IL-1RAcP. The active receptor triggers intracellular signaling cascades involving MyD88, IRAK, and MAPK to induce NF-kB- and AP-1-dependent expression of pro-inflammatory cytokines, chemokines, and secondary mediators of the inflammatory response. IL-36Ra and IL-38 bind to IL-36R inhibiting receptor dimerization and activation.

**Figure 3 ijms-20-01649-f003:**
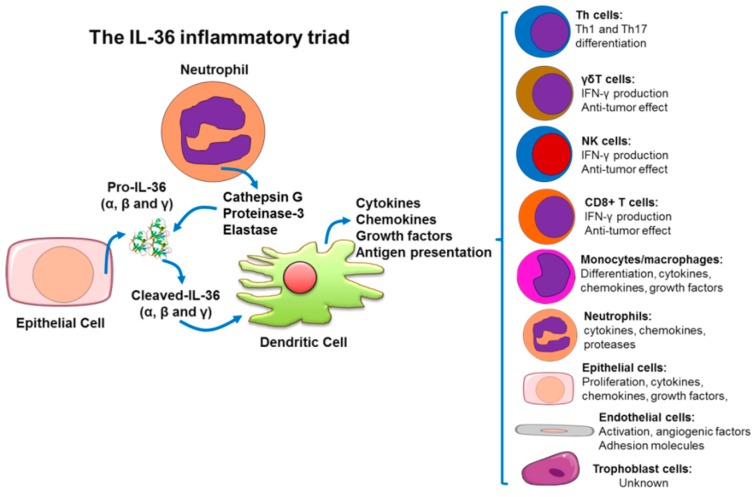
Origin and effects of IL-36 cytokines on different cell types. IL-36 cytokines are released by epithelial cells and processed by neutrophil-derived proteases to generate their active forms. The activated cytokines stimulate dendritic cells to enhance pro-inflammatory mediators responsible for several cellular effects in innate and adaptive immune responses. Some processes regulated on immune and non-immune cells are indicated. Note that there is still no data for trophoblast cells.

**Figure 4 ijms-20-01649-f004:**
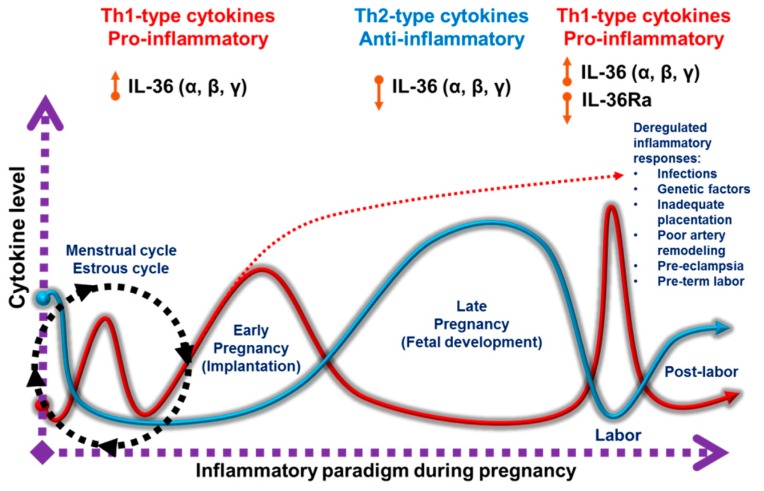
Illustrative description of the pro-inflammatory (red line) and anti-inflammatory (blue line) balance and IL-36 expression during pregnancy. Deregulated inflammatory reactions (red dotted line) may occur at any time in pregnancy and may overlap with each other. Depending on the stage, distinct complications can arise.

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
