# Peer review of "IL-36 Cytokines: Regulators of Inflammatory Responses and Their Emerging Role in Immunology of Reproduction"

_ijms, 2019, doi:10.3390/ijms20071649_

Round 1

Reviewer 1 Report

A great review of a cytokine family with emergent roles in pregnancy and I thoroughly enjoyed reading it. My comments are only minor and relate primarily to grammatical and typographical errors and some points for clarification.

General comments:

1.    The balance of this review is not in the favour of pregnancy so either a tweak to the title to reflect this or some key sentences related to the value of providing a detailed overview of IL-36 subfamily biology in the face of so little information in pregnancy is needed.

2.    Figure 1: it is not clear to me what the asterisk before IL-38 on the diagram denotes

3.    Section at line 253 needs an introductory section along the lines of “Insight into IL-36 family secretion strategies are only just emerging.’  

4.    At lines 271 – 274 some clarification is needed. What is the significance of substituting valine for methionine, what is this compared to?

5.    For figure 4: can data about IL-36Ra be added to this?

6.    Please review the many abbreviations used – they are familiar to me as an immunologist but might not be for other readers and might be better provided in the text rather than as a list. 

Minor points:

Line 25: remove ‘contributes to’

Line 44: ‘in’ should be ‘into’

Line 54: ‘a’ should be ‘an’

Line 55: ‘have developed and established’ should be ‘develop and establish’

Line 59: add mention of regulatory T cells and exosomes here as the examples you refer to

Lines 60: remove ‘from’

Lines 62/63: provide examples of these cells and factors

Line 69: add ‘and’ after [11];

Line 71: remove both commas

Line 72: remove ‘to’

Line 73: ‘leukocytes’ and ‘leucocytes’ used – choose one spelling to use throughout

Line 83: remove ‘contribute to’

Line 90: spell out ‘LPS

Line 90: ‘inhibitsa’ should be ‘inhibits a’

Line 95: remove ‘composed by’

Line 109: add ‘the’ before beginning

Line 138: 80’s should be 1980s

Line 138: ‘has been’ should be ‘was’

Line 139: remove ‘of the’ 

Line 139: remove second comma

Line 139: ’although’ should be ‘as’

Line 158: ‘alarmine’ should be ‘alarmin’

Line 159: remove ‘the’ at end of the line

Line 165: ‘releases a’ should be ‘release of’

Line 167: ‘a lower number of offsprinch, whihc’ should be ‘fewer offspring and these’

Line 169: ‘polyartropathy’ should be ‘polyarthropathy’

Line 171 (Figure legend): ‘constituted by’ should be ‘comprised of’

Line 175: add ‘a’ after with

Line 177: ‘implantats’ should be ‘implants’

Line 184: ‘being highest expressed’ should be ‘with highest expression’

Lines 188/189: ‘whose levels are even’ should be ‘with levels even’

Line 191: ‘has’ should be ‘is’

Line 192: ‘active form of 17.2kDa’ should be ‘active 17.2kDa form’

Line 205: ‘Contrarily’ should be ‘Contrary’

Line 205: add ‘estrus’ before ‘cycle’

Line 212: reference [70] should be at end of previous line

Line 214: ‘have been’ should be ‘were’

Line 217: remove comma

Line 217: remove ‘different’

Line 219: ‘has been’ should be ‘was’

Line 258: ‘See Figure 2’ versus ‘see figure 1’ at line 151 – be consistent  

Line 231: remove ‘are relevant for processes that’

Line 244: cell cultures of what cell type(s)?

Line 246: ‘lacks an N-glycosylation site in’ should be ‘lacks N-glycosylation at’

Line 249: ‘pro-dominium’ should be ‘pro-domain’

Line 250: ‘neither’ should be ‘any’

Line 250: ‘not know, yet’ should be ‘not yet known’

Lines 257/258: remove ‘cytokine release by blocking’

Line 260: ’neither’ and ‘nor’ should be ‘either’ and ‘or’

Lines 271 – 274 : Move ‘to IL-36a, IL-36b, and IL-36g’ to after mechanism

Line 288: add ‘family’ after IL-36

Lines 289/290: ‘cleaving’ should be ‘cleavage’

Line 295: remove ‘to’ 

Line 298: is there an additional space before IL-1RAcP?

Line 300: ‘has been’ should be ‘was’

Line 316: ‘maintaining’ should be ‘maintain’

Line 320: add ‘members’ after IL-36

Line 322: remove ‘of’

Line 327: ‘has’ should be ‘had’

Line 336: ‘ReIA’ should be ‘RelA’ – lower case l instead of capital I

Line 341: replace ‘the mediated’ with ‘their’

Line 345: should strips be sheets?

Lines 345/346: remove ‘pointed out as’

Line 360: hyphen should be after (IMQ)

Line 361: remove first ‘of’

Line 365: remove first ‘the’

Line 368: ‘of that’ should be ‘this’

Line 371: remove ‘are responsible to’

Line 379: ‘undifferentiation’ should be ‘dedifferentiation”; is this what you mean?

Line 386: remove ‘model’

Line 397: remove ‘receptors’

Line 401: ‘the lesioned’ should be ‘lesional’

Line 404: remove ‘explain’

Line 450: ‘them’ should be ‘these’

Line 455: ‘autocrine fashion as well as paracrinely’ should be ‘autocrine and paracrine fashion’

Line 459: remove ‘an’

Lines 460/461: ‘results to be critical in’ should be ‘contributes to’ or similar

Line 477: comma after obstruction

Line 480: what does TIL mean?

Line 499: ‘This’ should be ‘These’

Line 513: needs space before ‘In’

Line 516: remove ‘importangt’

Lines 527/528: where is IL-36 expressed in patients with Crohn’s disease?

Line 532: extra space before ‘M’ and ‘an’?

Line 535: ‘in vivowich’ should be ‘in vivowhich’

Line 539: should ‘treated’ be used instead of ‘stimulated’?

Line 544: replace ‘the’ with ‘do’

Line 544: ‘on’ should be ‘in’

Line 544: ‘This’ should be ‘These’

Line 545: ‘demonstrates’ should be ‘demonstrate’

Line 555: ‘producers and responders of’ should be ‘producers of and responders to’

Line 558: remove ‘to’

Line 570: need ‘are’ after ‘and’

Line 586: add ‘a’ before few

Line 586: ‘immune histochemical’ should be ‘immunohistochemical’

Line 589: ‘from’ should be ‘of’

Line 604: ‘enhanced’ should be ‘enhances’

Line 605: ‘role’ should be ‘roles’

Line 617: add ‘(DAMPs)’ after patterns

Line 625: ‘with’ should be ‘to’

Line 626: ‘pathogeny’ should be ‘pathogenesis’

Line 627: space needed after ‘of’

Line 630: space needed before ‘[‘

Line 634: space needed after ‘of’

Line 634: ‘expectable’ should be ‘expected’

Line 642: space needed before ‘in’

Author Response

Rebuttal letter

The authors appreciate all efforts dedicated to the revision of our manuscript. Suggestions made by the reviewers were integrally accepted. All modifications can be tracked down in the manuscript’s text. To better reflect the manuscript content, we have re-written the abstract. For the sake of clarification, we made a few minor additional changes in the text and included a sentence in lines 265-266:

“While cleavage of IL-1β and IL-18 by caspase-1 promotes their activation and secretion,…”.

All modifictions are highlighted. To avoid confusion, we did not update the line numbers provided by Reviewer 1 for very minor corrections. They still refer to the original manuscript. Requested modifications have been done as indicated and highlighted in the manuscript.

When sentences are reworded, we have updated line numbers.

Answers to Reviewer 1

A great review of a cytokine family with emergent roles in pregnancy and I thoroughly enjoyed reading it. My comments are only minor and relate primarily to grammatical and typographical errors and some points for clarification.

The authors thank the reviewer for the positive appreciation of our manuscript and for all suggested corrections.

General comments:

1.    The balance of this review is not in the favour of pregnancy so either a tweak to the title to reflect this or some key sentences related to the value of providing a detailed overview of IL-36 subfamily biology in the face of so little information in pregnancy is needed.

Both suggestions have been taken into consideration:

The title has been modified from “IL-36 – a family of cytokines and its potential involvement in the immunology of pregnancy” to “IL-36 cytokines: regulators of inflammatory responses and their emerging role in immunology of pregnancy”

The following sentence has been included in the text (lines 160-163): “Although information regarding the function of IL-36 cytokines in reproductive processes is still incipient, studies of different inflammatory conditions and immune responses illustrate how these cytokines may participate in regulating human reproduction”.

2.    Figure 1: it is not clear to me what the asterisk before IL-38 on the diagram denotes

The asterisk had been mistakenly added in the figure and has been removed.

3.    Section at line 253 (now line 275 in highlighted version) needs an introductory section along the lines of “Insight into IL-36 family secretion strategies are only just emerging.’  

The following sentence has been added: “Reports elucidating the molecular processing and secretory mechanisms employed by IL-36 cytokines have begun to emerge”.

4.    At lines 271 – 274 some clarification is needed. What is the significance of substituting valine for methionine, what is this compared to?

The paragraph has been modified as follows:

Lines 295-300 (in highlighted version): To acquire its antagonistic activity, IL-36Ra needs proteolytic processing. A methionine located at the N-terminus is removed leaving a valine at this position, which facilitates IL-36Ra interaction with IL-36R leading to its inhibition. Similar processes mediate the activation of IL‑1Ra and its binding to IL-1R. Due to this observation, it has been investigated whether a similar mechanism to IL-36α, IL-36β, and IL-36γ may be necessary to confer activity.

5.    For figure 4: can data about IL-36Ra be added to this?

Evidence indicates that IL-36Ra is decreased in complicated pregnancies. This information has been included in Figure 4.

6.    Please review the many abbreviations used – they are familiar to me as an immunologist but might not be for other readers and might be better provided in the text rather than as a list. 

            Abbreviations are now spelled in the text and a list.  

Minor points:

Line 25: remove ‘contributes to’

Corrected.

Line 44: ‘in’ should be ‘into’

Corrected.

Line 54: ‘a’ should be ‘an’

Corrected.

Line 55: ‘have developed and established’ should be ‘develop and establish’

Corrected.

Line 59: add mention of regulatory T cells and exosomes here as the examples you refer to:

Information regarding Tregs and exosomes has been included in lines 59-63:

“Trophoblast cells establish a cross-talk with T cells through soluble mediators and exosome, leading to the generation and activation of regulatory T cells (Tregs). Factors carried by extracellular vesicles “educate” resident and newly recruited Treg cells that suppress Th1 cell responses and favor fetal tolerance”.

Lines 60: remove ‘from’

            Corrected.

Lines 62/63: provide examples of these cells and factors

Lines 67-69: Cells and factors of fetal origin such as IL-1, TNF-α, MCP-1, MIP1-β, and IFN-γ modify maternal responses to stimuli [12, 13]. These fetal proinflammatory cytokines sensitize the pregnant mother to bacterial products and promote preterm labor [12]. 

Line 69: add ‘and’ after [11];

Corrected.

Line 71: remove both commas

Corrected.

Line 72: remove ‘to’

Corrected.

Line 73: ‘leukocytes’ and ‘leucocytes’ used – choose one spelling to use throughout

“leucocytes” has been substituted by “leukocytes” throughout the text.

Line 83: remove ‘contribute to’

Corrected.

Line 90: spell out ‘LPS

Done.

Line 90: ‘inhibitsa’ should be ‘inhibits a’

Corrected.

Line 95: remove ‘composed by’

Corrected.

Line 109: add ‘the’ before beginning

Corrected.

Line 138: 80’s should be 1980s

Corrected.

Line 138: ‘has been’ should be ‘was’

Corrected.

Line 139: remove ‘of the’ 

Corrected.

Line 139: remove second comma

Corrected.

Line 139: ’although’ should be ‘as’

Corrected.

Line 158: ‘alarmine’ should be ‘alarmin’

Corrected.

Line 159: remove ‘the’ at end of the line

Corrected.

Line 165: ‘releases a’ should be ‘release of’

Corrected.

Line 167: ‘a lower number of offsprinch, whihc’ should be ‘fewer offspring and these’

Corrected.

Line 169: ‘polyartropathy’ should be ‘polyarthropathy’

Corrected.

Line 171 (Figure legend): ‘constituted by’ should be ‘comprised of’

Corrected.

Line 175: add ‘a’ after with

Corrected.

Line 177: ‘implantats’ should be ‘implants’

Corrected.

Line 184: ‘being highest expressed’ should be ‘with highest expression’

Corrected.

Lines 188/189: ‘whose levels are even’ should be ‘with levels even’

Corrected.

Line 191: ‘has’ should be ‘is’

Corrected.

Line 192: ‘active form of 17.2kDa’ should be ‘active 17.2kDa form’

Corrected.

Line 205: ‘Contrarily’ should be ‘Contrary’

Corrected.

Line 205: add ‘estrus’ before ‘cycle’

“reproductive” has been added

Line 212:  [70] should be at end of previous line

Corrected.

Line 214: ‘have been’ should be ‘were’

Corrected.

Line 217: remove comma

Corrected.

Line 217: remove ‘different’

Corrected.

Line 219: ‘has been’ should be ‘was’

Corrected.

Line 258: ‘See Figure 2’ versus ‘see figure 1’ at line 151 – be consistent  

Corrected.

Line 231: remove ‘are relevant for processes that’

Corrected.

Line 244: cell cultures of what cell type(s)?

This information has been added in line 263: “Initial analyses of human epithelial and Jurkat T cell cultures revealed a high concentration of IL-36Ra…”

Line 246: ‘lacks an N-glycosylation site in’ should be ‘lacks N-glycosylation at’

Corrected.

Line 249: ‘pro-dominium’ should be ‘pro-domain’

Corrected.

Line 250: ‘neither’ should be ‘any’

Corrected.

Line 250: ‘not know, yet’ should be ‘not yet known’

Corrected.

Lines 257/258: remove ‘cytokine release by blocking’

Corrected.

Line 260: ’neither’ and ‘nor’ should be ‘either’ and ‘or’

Corrected.

Lines 271 – 274 : Move ‘to IL-36a, IL-36b, and IL-36g’ to after mechanism

Corrected.

Line 288: add ‘family’ after IL-36

Corrected.

Lines 289/290: ‘cleaving’ should be ‘cleavage’

Corrected.

Line 295: remove ‘to’ 

Modified to: “leading to the identification of ten members”

Line 298: is there an additional space before IL-1RAcP?

Corrected.

Line 300: ‘has been’ should be ‘was’

Corrected.

Line 316: ‘maintaining’ should be ‘maintain’

Corrected.

Line 320: add ‘members’ after IL-36

Corrected.

Line 322: remove ‘of’

Corrected.

Line 327: ‘has’ should be ‘had’

Corrected.

Line 336: ‘ReIA’ should be ‘RelA’ – lower case l instead of capital I

Corrected.

Line 341: replace ‘the mediated’ with ‘their’

Corrected.

Line 345: should strips be sheets?

Corrected.

Lines 345/346: remove ‘pointed out as’

Corrected.

Line 360: hyphen should be after (IMQ)

Corrected.

Line 361: remove first ‘of’

Corrected.

Line 365: remove first ‘the’

Corrected.

Line 368: ‘of that’ should be ‘this’

Corrected.

Line 371: remove ‘are responsible to’

Corrected.

Line 379: ‘undifferentiation’ should be ‘dedifferentiation”; is this what you mean?

The term has been substituted respectively.

Line 386: remove ‘model’

Corrected.

Line 397: remove ‘receptors’

Corrected.

Line 401: ‘the lesioned’ should be ‘lesional’

Corrected.

Line 404: remove ‘explain’

Corrected.

Line 450: ‘them’ should be ‘these’

            Corrected.

Line 455: ‘autocrine fashion as well as paracrinely’ should be ‘autocrine and paracrine fashion’

Corrected.

Line 459: remove ‘an’

Corrected.

Lines 460/461: ‘results to be critical in’ should be ‘contributes to’ or similar

Corrected.

Line 477: comma after obstruction

Corrected.

Line 480: what does TIL mean?

The abbreviation has been explained in line 508: “tubulointerstitial lesions (TIL)”.

Line 499: ‘This’ should be ‘These’

Corrected.

Line 513: needs space before ‘In’

Corrected.

Line 516: remove ‘importangt’

Removed.

Lines 527/528: where is IL-36 expressed in patients with Crohn’s disease?

Lines 562-564: The sentence has been re-written to include the requested information: “IL-36α and IL-36γ over-expression and IL-38 reduction have been found in the inflamed colon of a minor subpopulation of patients with Crohn’s disease”.

Line 532: extra space before ‘M’ and ‘an’?

Corrected.

Line 535: ‘in vivowich’ should be ‘in vivowhich’

Lines 570-572: The original sentence has been divided in two: “IL-36Ra decreases IL-6-driven inflammation as well as LPS- and IL-1β-induced MAPKs activity in hippocampal glial cells in vitro and in vivo. These responses are partially dependent on IL-4 induction and IL-36Ra activity in glia cells”.

Line 539: should ‘treated’ be used instead of ‘stimulated’?

Stimulated has been substituted by treated.

Line 544: replace ‘the’ with ‘do’

Corrected.

Line 544: ‘on’ should be ‘in’

Corrected.

Line 544: ‘This’ should be ‘These’

Corrected.

Line 545: ‘demonstrates’ should be ‘demonstrate’

Corrected.

Line 555: ‘producers and responders of’ should be ‘producers of and responders to’

Corrected.

Line 558: remove ‘to’

Corrected.

Line 570: need ‘are’ after ‘and’

Corrected.

Line 586: add ‘a’ before few

Corrected.

Line 586: ‘immune histochemical’ should be ‘immunohistochemical’

Corrected.

Line 589: ‘from’ should be ‘of’

Corrected.

Line 604: ‘enhanced’ should be ‘enhances’

Corrected.

Line 605: ‘role’ should be ‘roles’

Corrected.

Line 617: add ‘(DAMPs)’ after patterns

Corrected.

Line 625: ‘with’ should be ‘to’

Corrected.

Line 626: ‘pathogeny’ should be ‘pathogenesis’

Corrected.

Line 627: space needed after ‘of’

Corrected.

Line 630: space needed before ‘[‘

Corrected.

Line 634: space needed after ‘of’

Corrected.

Line 634: ‘expectable’ should be ‘expected’

The sentence was modified to avoid repetition of “expected”, already present in the previous sentence.

Line 642: space needed before ‘in’

Corrected.

Reviewer 2 Report

I read with great interest the Manuscript titled “IL-36 – a family of cytokines and its potential involvement in the immunology of pregnancy” (ijms-469304).

In my honest opinion, the topic is interesting enough to attract the readers’ attention. Writing is accurate and conclusions are supported by the literature data. Nevertheless, authors should clarify some points and improve the discussion citing relevant and novel key articles about the topic.

Authors should consider the following recommendations:

- Manuscript should be further revised by a native English speaker.

- Authors have appropriately addressed how cytokine network plays a key role in modulating immune response at the maternal-fetal interface. Nevertheless, I would suggest to stress how these fine-regulated machinery may be influenced selectively by epigenetic changes, such as miRNAs expression. Refer to: PMID: 28466013; PMID: 28282763.

- Although pro-inflammatory cytokines are known to regulate the correct remodelling of arteriae spiralis during embryo implantation, accumulating evidence suggests that both Endothelial Progenitor Cells (EPCs) and Natural Killer (NK) may have a strong influence about this mechanism. Authors should discuss how the disbalance between them may alter the process of placental development and generate the immune background for both pre-eclampsia and fetal growth restriction, referring to: PMID: 28243732; PMID: 29923045

Author Response

Answers to Reviewer 2

Read with great interest the Manuscript titled “IL-36 – a family of cytokines and its potential involvement in the immunology of pregnancy” (ijms-469304).

In my honest opinion, the topic is interesting enough to attract the readers’ attention. Writing is accurate and conclusions are supported by the literature data. Nevertheless, authors should clarify some points and improve the discussion citing relevant and novel key articles about the topic.

The authors thank the reviewer for the comments.

Authors should consider the following recommendations:

- Manuscript should be further revised by a native English speaker.

We have revised the language upon the detailed corrections as requested by Reviewer 1.

- Authors have appropriately addressed how cytokine network plays a key role in modulating immune response at the maternal-fetal interface. Nevertheless, I would suggest to stress how these fine-regulated machinery may be influenced selectively by epigenetic changes, such as miRNAs expression. Refer to: PMID: 28466013; PMID: 28282763.

These references have been added in lines 132-136 as follows:

“Several regulatory mechanisms control the balance between pro-inflammatory and anti-inflammatory reactions at the maternofetal interface. Epigenetic regulation of DNA methylation, imprinting and microRNAs (miRNAs) play pivotal roles on these processes. miRNAs are involved in placental development and changes on their expression are related with placental disorders and pregnancy complications [9, 45-47]”.

- Although pro-inflammatory cytokines are known to regulate the correct remodelling of arteriae spiralis during embryo implantation, accumulating evidence suggests that both Endothelial Progenitor Cells (EPCs) and Natural Killer (NK) may have a strong influence about this mechanism. Authors should discuss how the disbalance between them may alter the process of placental development and generate the immune background for both pre-eclampsia and fetal growth restriction, referring to: PMID: 28243732; PMID: 2992304

One reference has been added in lines 88-89:

“An imbalance in the number and functionality of uterine NK cells and endothelial progenitors together with changes in pro- and anti-angiogenic factors contributes to preeclampsia onset [28]”.
